# Experimental Characterization of Polymer Surfaces Subject to Corona Discharges in Controlled Atmospheres

**DOI:** 10.3390/polym11101646

**Published:** 2019-10-10

**Authors:** Andres R. Leon-Garzon, Giovanni Dotelli, Matteo Tommasini, Claudia L. Bianchi, Carlo Pirola, Andrea Villa, Andrea Lucotti, Benedetta Sacchi, Luca Barbieri

**Affiliations:** 1Dipartimento di Chimica, Materiali e Ingegneria Chimica “Giulio Natta”, Politecnico di Milano, 20133 Milano, Italy; andresricardo.leon@polimi.it (A.R.L.-G.); matteo.tommasini@polimi.it (M.T.); andrea.lucotti@polimi.it (A.L.); 2Dipartimento di Chimica, Università degli Studi di Milano, 20133 Milano, Italy; claudia.bianchi@unimi.it (C.L.B.); carlo.pirola@unimi.it (C.P.); benedetta.sacchi@unimi.it (B.S.); 3Ricerca sul Sistema Energetico (RSE S.p.A.), 20134 Milano, Italy; andrea.villa@rse-web.it

**Keywords:** partial discharges, low-temperature plasma, polymer surface degradation, chemical characterization, morphological characterization

## Abstract

Polymeric dielectrics are employed extensively in the power transmission industry, thanks to their excellent properties; however, under normal operating conditions these materials tend to degrade and fail. In this study, samples of low-density polyethylene, polypropylene, polymethyl methacrylate, and polytetrafluorethylene were subjected to corona discharges under nitrogen and air atmospheres. The discharges introduced structural modifications over the polymer surface. From a chemical perspective, the alterations are analogous among the non-fluorinated polymers (i.e., polyethylene (PE), polypropylene (PP), and polymethyl methacrylate (PMMA)). A simulation of the corona discharge allowed the identification of highly reactive species in the proximity of the surface. The results are consistent with the degradation of insulating polymers in high-voltage applications due to internal partial discharges that ultimately lead to the breakdown of the material.

## 1. Introduction

Polymeric materials have been extensively employed in the electrical power industry as insulators due to their remarkable and wide-ranging properties [1]. In particular, great attention has been paid to the use of polyethylene (PE) in high-voltage cables in the last decades [2]. Despite this, under normal operation conditions, the polymer is subject to several stresses (e.g., mechanical, thermal, electrical, chemical, etc.) that lead to its degradation or aging and, in some cases, to the final failure of the insulator.

One particular phenomenon that has proven to be considerably detrimental is the occurrence of partial discharges (PDs) [3,4,5]. Partial discharges are electrical discharges that do not involve any direct bridge between the electrodes and they occur due to the electrical breakdown of a low-density media in close contact with the polymer matrix [6,7]. They can be classified into four types: corona discharges, surface discharges, internal discharges, and electrical treeing [8].

It is generally accepted that the continuous bombardment over the polymer surface by the active species, produced in the discharge, destroys the matrix creating pits that eventually evolve into channels in a phenomenon called electrical treeing. The high electric field triggers several reactions between the main constituents of the low-density media (in actual HV applications, it is typically air), which in turn creates the active species that participate in the degradation of the polymer matrix [3,9]. Prolonged deterioration by PD will provoke the deposition of carbonaceous compounds in the wall of the channels leading to a conducting tree that will cause the ultimate dielectric failure [10,11].

There are several examples of scientific literature on the behavior, measurement, and diagnostics of partial discharges [12,13,14,15,16,17,18,19]; however, the main chemical mechanism that contributes to the propagation of partial discharges and electrical treeing is still largely unknown.

For instance, in Morshuis review, a selection of studies were gathered dealing with the degradation of different polymers under partial discharges and corona discharges [20]. It was shown that the main gaseous degradation products are carbon monoxide, carbon dioxide, hydrogen, and methane. On the other hand, in the solid phase, it is suggested that sustained discharges will eventually lead to the formation of hydrated crystals of oxalic acid (C_2_H_2_O_4_·H_2_O).

More recently, Sekii studied the degradation of low-density polyethylene (LDPE) and cross-linked polyethylene XLPE by partial discharges employing a spherical-plane electrode system in different atmospheres [21]. A qualitative evaluation of the gases developed by the discharge was also made. It was found once more that carbon dioxide is the most prominent gas developed when employing an oxygen-containing atmosphere. Employing infrared spectroscopic analysis, the author evidenced the formation of carbonyl groups (–C=O) on the surface of the polymer.

Adhikari, Hepburn, and Stewart reproduced a series of vented and unvented self-contained voids using multiple layers of polyethylene terephthalate (PET) [22]. Using chemical and morphological analyses, they found that oxidation in unvented voids is higher than in vented voids, as in the latter case the partial discharge by-products are able to disperse and the gas can be refreshed through the vent.

On the other hand, the available literature on the study of the effects of plasma treatments on polymers is rather large. For instance, studies on the modification of the surface properties of polymeric materials in several fields have been performed employing dielectric barrier discharges, plasma jets, glow discharges [23,24,25,26,27,28,29,30,31,32,33], and plasma polymerization [34,35,36]. Despite this, the operating conditions (discharge power, pressure, etc.) of these plasma treatments are far from the ones present in a partial discharge.

As a result, in this work, we present the results of the study of the effects of corona discharges over polymeric materials employing experimental techniques to determine structural and chemical modifications. The study of such effects was done employing slab samples of low-density polyethylene (PE), polypropylene (PP), polymethyl methacrylate (PMMA), and polytetrafluorethylene (PTFE). We acknowledge that, in many practical cases, PE is the most common choice in power industry; however, PP and PMMA, since they are transparent, or semi-transparent, are used in a number of articles [37]. Moreover, as we will show, PTFE, in spite of its higher weight and cost, has some better insulation performances. The polymer slabs were subjected to corona discharges in both air and nitrogen atmospheres. The structural modifications induced by the corona discharges were analyzed employing scanning electron microscopy (SEM). Finally, Fourier transform infrared spectroscopy (FTIR) and X-ray photoelectron spectroscopy (XPS) were employed to qualitatively characterize the chemical changes over the polymers surface.

The modifications observed can be employed to extrapolate the degrading effects of internal partial discharges and to determine how generic the modifications are with respect to the composition of the polymer. Similarly, such observations can help individualize those factors that cause the greatest modification on the material and to outline a primary chemical mechanism of PD degradation in polymeric dielectrics. For this purpose, we have also developed a 3D simulation of the experimental corona discharge system to identify those species with the highest probability of modifying the polymer surface. This information will be employed on a future work on the modeling of the degradation phenomenon of polymeric materials, and in particular polyethylene, subject to partial discharges.

This paper is structured as follows: the experimental setup and the material characterization techniques employed in this work are described in Section 2, the experimental and modeling results are presented and discussed in Section 3. Finally, in Section 4, we present the conclusions.

## 2. Materials and Methods

### 2.1. Materials

Polymer slabs (50 mm × 50 mm, Goodfellow, Huntingdon, UK) of low-density polyethylene, polypropylene, polymethyl methacrylate, and polytetrafluoroethylene were subjected to corona discharges to study their effects on the material surface. The discharges were carried out in controlled atmospheres of high purity air ALPHAGAZ 1, (Air Liquide, Paris, France), O_2_ 20% ± 1%, N_2_ 80% ± 1%) and nitrogen (Air Liquide ALPHAGAZ 1, N_2_ 99.999%).

### 2.2. Experimental Setup

The experimental setup consists of a system of rod-plane electrodes as described in Figure 1. The high-voltage electrode is a stainless-steel rod with hollow tip in its end. The thickness of the tip’s wall is of 1 mm, and it is placed at 2.3 mm above the surface of the polymer slab. The size of the gap between the slab and the rod has been chosen as a compromise to get a rather homogeneous treatment of the central area of the sample. In fact, if we had placed the rod just in contact, then no surface discharge would have taken place in the central part of the needle cone. Otherwise, if the distance had been too great, then the chemical radical active species would have recombined in the gas phase without reaching the plastic specimen. Due to the reduced size of the polymer sample, the same is placed over a PMMA slab that acts as dielectric barrier, which prevents spark or arc formation. The PMMA slab is in turn supported by a circular stainless-steel plane that constitutes the bottom of a spherical stainless-steel chamber that shelters the system. Such chamber is isolated from the high-voltage electrode and connected to the ground potential and can be evacuated and filled with the preferred gas. An AC voltage (50 Hz) applied to the bar electrode and increased until a partial discharge event was recorded. The partial discharges were recorded employing an OMICRON MD (Omicron, Klaus, Austria) 600, an optically isolated IEC 60270 compliant partial discharge measurement device. The discharge was sustained taking care that the pulses did not exceed more than 5 nC to limit the regime of the discharge. Finally, the polymer samples were treated for 72 h to maximize the surface modifications and were subsequently stored under ambient atmosphere.

### 2.3. Material Characterization

#### 2.3.1. Scanning Electron Microscopy (SEM)

The morphological modifications were analyzed with a Cambridge (Cambridge Stereoscan, Cambridge, UK) Stereoscan 360 Scanning Electron Microscopy (SEM). Since the conductivity of the polymers is low, the samples were prepared previously by sputter coating with an ultra-thin gold layer. It is worth noting that, due to the intrusion of the gold layer, the SEM characterization was done after both the FTIR and XPS analyses.

#### 2.3.2. Fourier Transform Infrared Spectroscopy (FTIR)

The chemical modifications of the polymer surfaces can be qualitatively determined by infrared spectroscopy. The attenuated total reflectance (ATR) technique was employed to analyze the surface of the samples (to a depth between 0.5 and 2 µm) without the need of previous preparation.

The FTIR-ATR spectra were collected using a Nicolet (Nicolet Instrument, Madison, WI, USA) Nexus instrument coupled with an infrared microscope Thermo (Thermo Fisher Scientific, Waltham, MA, USA) Scientific Nicolet Continuum equipped with an MCT detector cooled to 77 K. The ATR attachment is a single bounce Thermo Scientific Slide-On objective made with a hemispherical silicon tip; this objective has a refraction index of 3.4, an angle of incidence of 55°, and a typical sample area of 25 µm. The collection of the spectra was made employing 32 scans in the 650–4000 cm^−1^ spectral range with a spectral resolution of 4 cm^−1^.

#### 2.3.3. X-ray Photoelectron Spectroscopy (XPS)

XPS spectra were obtained using a Surface Science Instruments (Surface Science Instruments, Mountain View, CA, USA) M-probe apparatus. The X-ray source was an Al−K α monochromatic beam (1486.6 eV). In survey analysis, the spot size is equal to 800 μm with a resolution of 4 eV, while in high resolution analysis the spot size is equal to 150 μm with a resolution of 1 eV.

## 3. Results and Discussion

### 3.1. SEM

#### 3.1.1. Polyethylene

The images detailing untreated and treated surfaces of PE are shown in Figure 2. Untreated polyethylene presents a relatively uniform surface with some roughness due to the mechanical abrasion already present in the polymer slab. In contrast, a dispersion of granular-like formations is clearly visible on the slab surface after discharge under both air and nitrogen atmosphere. These granular-like formations are slightly bigger and more dispersed in the air treatment. Similar formations have also been observed after the degradation of polyethylene terephthalate by partial discharges in unvented voids [22].

#### 3.1.2. Polypropylene

SEM images of untreated, air-treated and nitrogen-treated surfaces of PP are shown in Figure 3. In the air-treated polymer there is an accumulation of jagged protrusions over the surface. This could be related to an increase on the surface roughness of PP observed by Pandiyaraj et al. who employed a glow discharge treatment [33]. On the other hand, the morphological features of the nitrogen treated PP surfaces share some characteristics with the PE treated surfaces as there is a formation of granular-like protrusions on the surface. In the present case, however, the granules are less abundant and more dispersed. It is worth noting that these formations have also been observed after the treatment of PP surfaces by atmospheric pressure nitrogen glow discharges [32].

#### 3.1.3. Polymethyl Methacrylate

Surface images of treated and untreated PMMA are shown in Figure 4. The magnification was limited to a value of 2500× to avoid radiation damage on the polymer surface by the incident electron beam. In both cases, an increase of surface roughness is expected due to the formation of particulate-like matter. Similar results have been obtained in PMMA surfaces treated by atmospheric dielectric barrier discharges (DBD) [38]. From the SEM images, the roughening effect is more prominent in nitrogen than in air atmosphere. This comes from the fact that nitrogen plasma treatments introduce large changes in the surface morphology even though the etching rate is slower than in oxygen plasma treatments [39].

#### 3.1.4. Polytetrafluoroethylene

Images of untreated and treated PTFE surfaces are shown in Figure 5. The morphological behavior of PTFE surfaces differs greatly from the previous cases as it is possible to appreciate that discharges in air introduce rod-shaped formations on the surface. On the other hand, nitrogen discharges create comparable formations that are much smaller and less widespread. Similar results have been observed previously as Fang et al. noted the formation of protrusions on PTFE surfaces after treatment with atmospheric DBD, while Sarra-Bournet et al. detected almost no modification on the fluorinated polymer when analyzing the effects of nitrogen DBD discharges [40,41].

### 3.2. FTIR

The infrared spectra of the treated and untreated polymer surfaces are shown in Figure 6. In each panel, the untreated spectrum corresponds to the bottom-most curve (depicted in green), while the air-treated and the nitrogen-treated spectra are, respectively, the middle and the top-most curves (depicted correspondingly in orange and violet).

#### 3.2.1. Untreated Polymers

The fundamental frequencies of untreated polyethylene active in infrared spectra are essentially the modes of vibration of the −CH_2_− group. First, the rocking mode ρ(C−H) gives an intense band around 720 cm^−1^. Second, the bending mode δ(C−H) is active around 1470 cm^−1^. Finally, the symmetric ν_s_(C−H) and antisymmetric ν_a_(C−H) stretching modes are present around 2850 cm^−1^ and 2917 cm^−1^, respectively [42].

Untreated polypropylene gives rise to two bands around 1375 cm^−1^ and 1460 cm^−1^. The former corresponds to the symmetric deformation of −CH_3_ groups while the latter corresponds to the bending of C−H bonds in −CH_2_− groups and/or the asymmetric deformation of −CH_3_ groups. The peaks pertaining to the 2840 cm^−1^ and 2918 cm^−1^ wavenumber correspond, respectively, to the symmetric and asymmetric stretching modes ν(C−H) in the −CH_2_− groups of the backbone of the polymer chain. Finally, the two remaining bands pertaining to the 2870 cm^−1^ and 2955 cm^−1^ wavenumber are attributed, respectively, to the symmetric and asymmetric stretching mode ν(C−H) in −CH_3_ groups [26].

The most distinctive characteristics of the untreated polymethyl methacrylate spectrum are the peaks from 1150 cm^−1^ to 1240 cm^−1^ that are attributed to the ν (C−O−C) stretching vibrations. The peak around the 1453 cm^−1^ wavenumber corresponds to the bending vibrations of δ(C−H) bonds in −CH_3_ groups. The band around 1725 cm^−1^ corresponds to the stretching vibrations ν(C=O) in carbonyl bonds. Finally, the bands around 2950 cm^−1^ and 2994 cm^−1^ correspond to the stretching vibrations of ν(C−H) in −CH_2_− groups and −CH_3_ groups, respectively [38].

Finally, the IR spectrum of untreated polytetrafluoroethylene presents two peaks, the first around 1205 cm^−1^ and the second at 1150 cm^−1^ that correspond, respectively, to the symmetric ν_s_(C−F) and asymmetric ν_a_(C−F) stretching in −CF_2_− groups [43].

#### 3.2.2. Treated Polymers

The modifications due to the corona discharge appear mainly in three particular regions of the infrared spectra in the case of PE, PP, and PMMA. We have identified these three regions using the labels A, B, and C as seen in Figure 6. In region A, which corresponds to the 3400–3050 cm^−1^ wavenumber, a rather broad contribution can be observed after the treatment of PE, PP, and PMMA with air and nitrogen corona discharges. This can be attributed to the stretching vibrations ν(O−H) in hydroxyl groups. The broadness of this contribution is a marker of the presence of hydrogen bonds [42]. Region B corresponds to the 1800–1550 cm^−1^ wavenumber; here, a weak shoulder can be observed, which can be attributed to the stretching vibration ν(C=O) in carbonyl groups with a different chemical environment [42]. Finally, in region C, which is extended between the 1450–1300 cm^−1^ wavenumber, presents a contribution that can be attributed to the bending vibrations δ(O−H) in hydroxyl groups; however, if the formation of low molecular weight (LMW) fragments is involved, the contribution by the bending vibrations δ(C−H) in −CH_2_− or −CH_3_ groups cannot be discarded [44].

The contribution of the stretching vibrations ν(N−H) in region A and the stretching vibration ν(C=N) in region B cannot be rejected. Nitrogen discharges can be quite effective incorporating N atoms in the surface, thus, explaining the slightly higher signal in the nitrogen-treated samples [45]. However, as will be illustrated later, the surface chemistry is highly influenced by the presence of oxygenated compounds during and after the discharge, and the stability and further availability of nitrogen groups in the surface is greatly compromised. Lastly, the corona treatment in either air or nitrogen over PTFE surfaces did not introduce any kind of modification to the polymer, thanks to the stability of the carbon–fluorine bonds [28].

### 3.3. XPS

The atomic concentration of carbon, oxygen, and nitrogen atoms by XPS analysis can be seen in Table 1. In this analysis, we have treated the PE and PP cases since PMMA has already oxygen-based functional groups in its structure that will make difficult the observance and assignment of new contributions due to the discharge. On the other hand, as seen in the FTIR, PTFE was not affected by any significant chemical modifications after the treatment. Both PE and PP, in their untreated form, contain oxygen in the surface, which can be attributed to a low-level oxidation of the polymer melt when passing through the extruder [26,46]. However, after the treatment of PE with nitrogen and air corona discharges, the oxygen content increases from 7.53 to 10.68% and from 7.53 to 16.06%, respectively. Similarly, the oxygen content in PP increases from 6.86 to 11.24% in the nitrogen-treated sample and from 6.86 to 10.62 % in the air-treated sample. In both PE and PP, the nitrogen content is below 4% after the treatment with nitrogen corona discharges and under 1% in the air-treated samples. Additionally, it can be observed in Table 1 that, even under a nitrogen treatment, there is a higher concentration of oxygen atoms with respect to the nitrogen content. This phenomenon can be explained in two ways. First, even in the presence of small quantities of oxygen (i.e., ppm) in the atmosphere, the chemistry of the surface discharge changes radically as the incorporation of nitrogen atoms is replaced by oxygen atoms [45,47]. Second, some of the nitrogen containing groups, after being exposed to air, can be replaced with oxygen-containing groups and/or create some new groups that solely contain oxygen atoms [45,48,49].

The functionalities introduced by the corona discharges were further analyzed employing high resolution XPS analysis of the carbon (C1s) and oxygen (O1s) peaks. The contributions of the functionalities were calculated by deconvolution using a Gaussian–Lorentzian fit. The deconvolutions of the C1s peak of the nitrogen-treated and air-treated PE and PP are shown in Figure 7. The C1s peaks are fitted employing three peaks noted as C1, C2, and C3. Peak C1 (284.6 eV) can be attributed to aliphatic carbons C−C/C−H [50,51]. The second peak (C2) at 285.4 eV may be attributed to C−O bonds and in some extent to C−N bonds [52,53,54]. Finally, peak C3 (288 eV) can be assigned to carbonyl and acetal groups (C=O and O−C−O) [46,55].

Figure 8 shows the deconvolution of the O1s peaks of nitrogen-treated and air-treated PE and PP. Two peaks, labeled O1 and O2, are employed to fit the O1s spectrum. Peak O1, centered on 532 eV, can be attributed to the presence of C=O and O−C−O groups. Meanwhile, peak O2 (533.3 eV) can be assigned to C−O groups [56,57].

Consequently, oxygen thus play a major role in the degradation of polymers under partial discharges while, on the other hand, nitrogen-based species are less prominent since their formation is impeded during and after the treatment. The post-treatment degradation in nitrogen-treated samples is presumably due to their exposition with atmospheric air, which triggers the reactions [58]:(1)[C=NH+H2O(g)⇒[C=O+NH3,
(2)[C=N−R+H2O(g)⇒[C=O+H2N−R,
(3)[C=NH2+H2O(g)⇒[C−OH+NH3.

As expected, the structural and chemical modifications are comparable in the three non-fluorinated polymers employed in the present work. In a similar way, it is interesting to notice that the degradation by internal partial discharge within self-enclosed voids in polyethylene terephthalate (PET) reveals aspects that are analogous to the present findings. As a result, similar granular-like structures were found in the SEM analysis of internal PD-degraded PET, while the infrared analysis revealed new contributions due to O−H bonds and a new weak band associated to C=O bonds which are in agreement with the results obtained for non-fluorinated polymers [22]. As a result, the chemical modifications introduced by partial discharges share a common mechanism of degradation, which is mostly associated with the oxidation of the material.

### 3.4. Modelling of the Corona Discharge

As seen in the experimental result, the presence of oxygen is crucial in the degradation of polymers under corona discharges. Therefore, it is expected that some of the oxygen-based species generated by the discharge will play a major role in the degradation of polymeric insulators subject to partial discharges as well.

We have developed a simulation of the experimental setup in Figure 1 for a discharge under air atmosphere. The model is based on numerical techniques previously developed for the simulation of corona discharges [59,60,61,62,63]. The model is based on a set of partial differential equations for the evolution of the concentrations of charged and neutral species coupled with the electrostatic equation. The charged species evolve according to a set of drift diffusion equations with reaction terms that come from plasma kinetics. On the other hand, the neutral particles evolve according to the Euler equations of gas dynamics. The computational domain is depicted in Figure 9. We have employed a reaction scheme for humid air introduced by Sakiyama [64]. The scheme is composed by more than 600 reactions and includes 53 species among electrons, positive ions, negative ions, and neutral species. The swarm parameters were calculated employing METHES while the cross-sections were obtained from the LXCat project website [65,66]. Further details about the scheme and the techniques employed for modeling of corona discharges can be found in Villa et al. [63]. The initial conditions of the simulation are *T* = 300 K, *P* = 101,000 Pa, $y_{N_{2}}$ = 0.79, $y_{O_{2}}$ = 0.2 and $y_{H_{2}O}$ = 0.01. The concentrations of both electrons (*n_e_*) and positive ions (*n_p_*) were set to 10^10^ m^−3^, while that of the negative ions (*n_n_*) was set to zero. Finally, the AC potential (50 Hz) applied to the rod electrode is equal to 12.8 kV.

The evolution of the electric charge within the volume is shown in the top panel of Figure 10. As expected, the discharge generates a series of pulses as electrons are multiplied due to the ionization of the molecular constituents of air. The discharge does not only generate electrons and ions, but collisions between molecules and electrons with energies below the ionization threshold (i.e., 15.58 eV for N_2_, 12.07 eV for O_2_, and 12.6 eV for H_2_O) are also able to generate several other neutral species.

In Table 2 are listed the relative peak concentrations of selected most abundant neutral species generated at each of the pulses (without including N_2_, O_2_, and H_2_O which are predominant in the system). The concentrations were measured in correspondence of a point along the domain’s cylindrical axis that lies just over the polymer surface (red dot in Figure 9). The decay time of the O(^3^P) and OH radicals depends both on the surface-gas chemical reaction and on reactions just in the gas phase. In particular, O(^3^P) quickly reacts with other species of gas and then interacts with the surface. On the contrary, OH reacts more slowly in the gas and therefore it decays more slowly.

Of these species, atomic oxygen (O(^3^P)) and the hydroxyl radical (OH) are mostly considered to be responsible for the initiation of the degradation process by the abstraction of a hydrogen atom from the polymeric chain [67]. In polyethylene, this can be written as(4)[CH2−CH2−CH2]+O(3P)⇒[CH2−CH−CH2]+OH,
(5)[CH2−CH2−CH2]+OH⇒[CH2−CH−CH2]+H2O.

Other species might also contribute to the degradation of the polymer. For example, ozone (O_3_) is known to accelerate the aging polymeric materials; however, its reactions take place predominantly through an ozonolysis mechanism with unsaturated groups that can be considered absent in polyethylene [68]. On the other hand, atomic nitrogen (N) is also known to attack polyolefins but its mechanism is not well known and the literature on this topic is scarce [47].

The evolution of the atomic oxygen and hydroxyl radical concentrations measured are shown in the middle and bottom panel of Figure 10. In a discharge in humid air, atomic oxygen is generated by the electron-impact dissociation of molecular oxygen.(6)$$e^{-}+O_{2}\Rightarrow 2O\left({}^{3}P\right)+e^{-}.$$

Similarly, water is dissociated into OH and H radicals by the reaction(7)$$e^{-}+H_{2}O\Rightarrow \mathrm{OH}+H+e^{-}.$$

There is rapid growth in the concentration of these species in correspondence of the current pulses. In the case of atomic oxygen, the rapid growth is followed by an almost instant decrease of the concentration generating a series of concentration peaks. On the other hand, the concentration of the hydroxyl radical decreases significantly slower. The time-average O/OH ratio is about 2.23 indicating that the polymer degradation will be dominated by the oxygen atom, which is typical of corona discharges [47,69]. The reason for this is twofold. Firstly, although the reaction rate of the H-abstraction by OH is one order of magnitude greater than that of O(^3^P) ($k_{\mathrm{OH}}/k_{O}\approx 50$) [70], the reaction of triplet oxygen with the polymer surface will undoubtedly generate an OH radical that will further deteriorate the material. Secondly, the generation of OH radicals is significantly affected by the concentration of water in air (i.e., through the reaction in Equation (7)). Thus, in a lower humidity environment, the importance of OH radicals in the polymer degradation will be less significant.

## 4. Conclusions

Surface discharges over samples of low-density polyethylene (PE), polypropylene (PP), polymethyl methacrylate (PMMA), and polytetrafluoroethylene (PTFE) under air and nitrogen atmospheres were carried out employing a bar-plate system. The main findings obtained can be summarized as follows:From a structural point of view, both discharges in nitrogen and air atmosphere were able to modify to a large extent the surface of the PE, PP, and PMMA samples. In both atmospheres, polymer surfaces suffered from oxidation contributions since hydroxyl and/or carbonyl functional groups were observed by FTIR and XPS; such modifications closely resemble those obtained by partial discharges in unvented voids embedded in PET [22].No functionalities were observed on the PTFE surfaces due to the significant stability of C–F bonds. On the contrary, the modifications introduced by corona discharges over non-fluorinated polymer surfaces are comparable with each other and with those obtained by partial discharge degradation on XLPE and PET [21,22].3D simulation of the system confirmed the availability of oxygen-based species, and in particular of atomic oxygen, in close proximity to the polymer surface. These species are readily active to cause chemical modifications to the material.

Comparing these findings with the existing literature, we can state that:our results are analogous to the modifications observed after the degradation of polyethylene terephthalate due to partial discharges in enclosed voids but using rather different experimental conditions;this suggests that the aging of the specimens share a common degradation mechanism from a chemical point of view.

The production of polyethylene employs several processes such as extrusion, molding, vulcanization, etc. These processes can introduce micro-sized cavities within the polymer matrix and since most of them are carried out under air atmosphere, the presence of oxygen in such microcavities cannot be discarded. For instance, the electrical breakdown of the air contained within such microcavities will cause internal partial discharge that eventually leads to electrical treeing and to the failure of the dielectric. The modeling of such degradation phenomenon is directly linked to what is discussed in this work and it will be a central part of our future work.

## Figures and Tables

**Figure 1 polymers-11-01646-f001:**
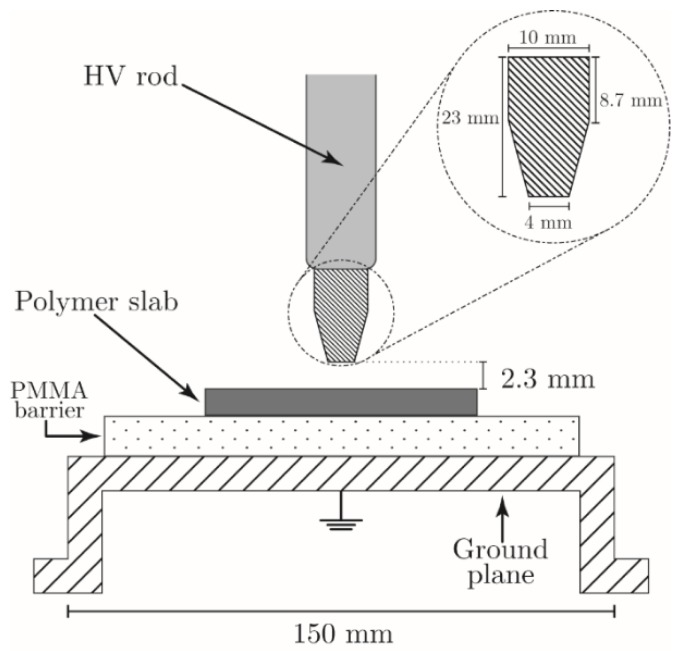
Experimental setup.

**Figure 2 polymers-11-01646-f002:**
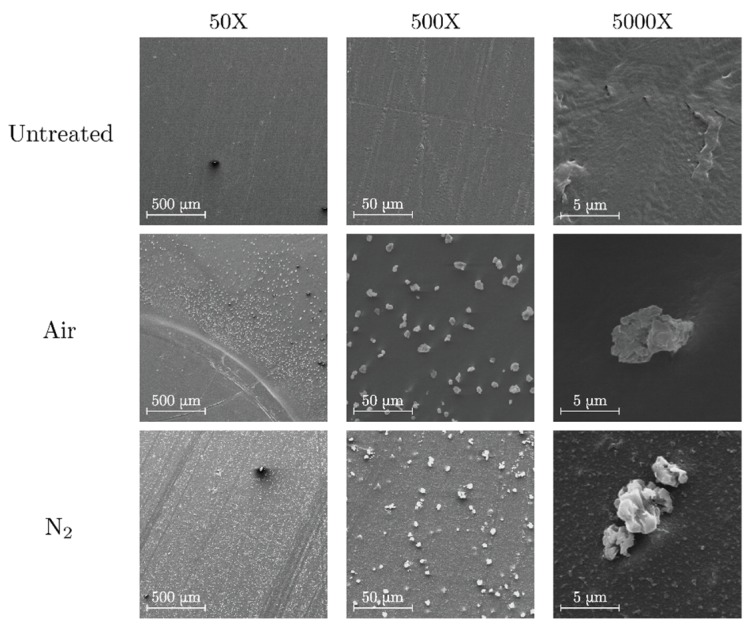
SEM images of untreated (top), air-treated (center) and nitrogen-treated (bottom) polyethylene (PE).

**Figure 3 polymers-11-01646-f003:**
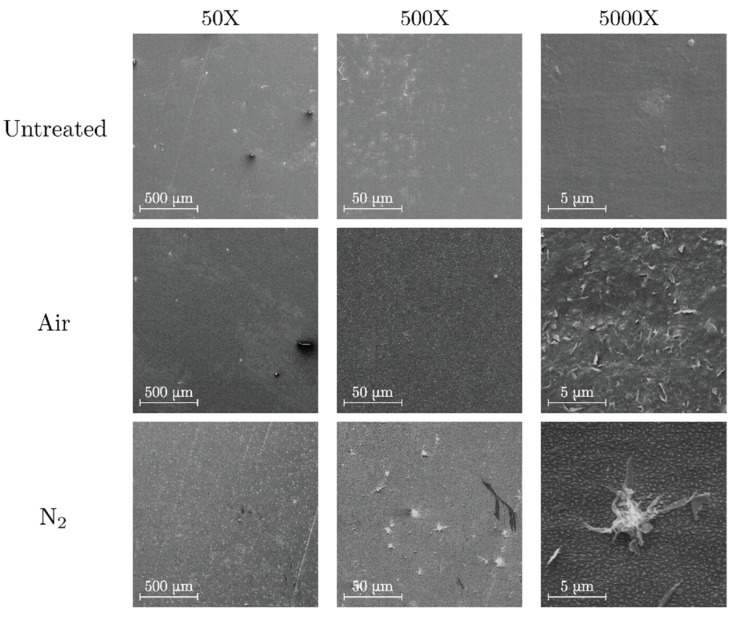
SEM images of untreated (**top**), air-treated (**center**) and nitrogen-treated (**bottom**) PP.

**Figure 4 polymers-11-01646-f004:**
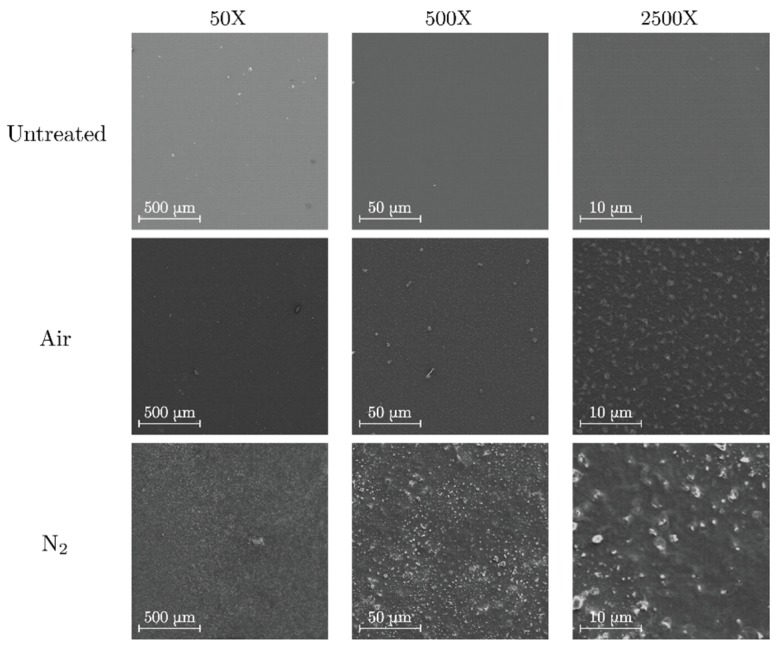
SEM images of untreated (**top**), air-treated (**center**) and nitrogen-treated (**bottom**) polymethyl methacrylate (PMMA).

**Figure 5 polymers-11-01646-f005:**
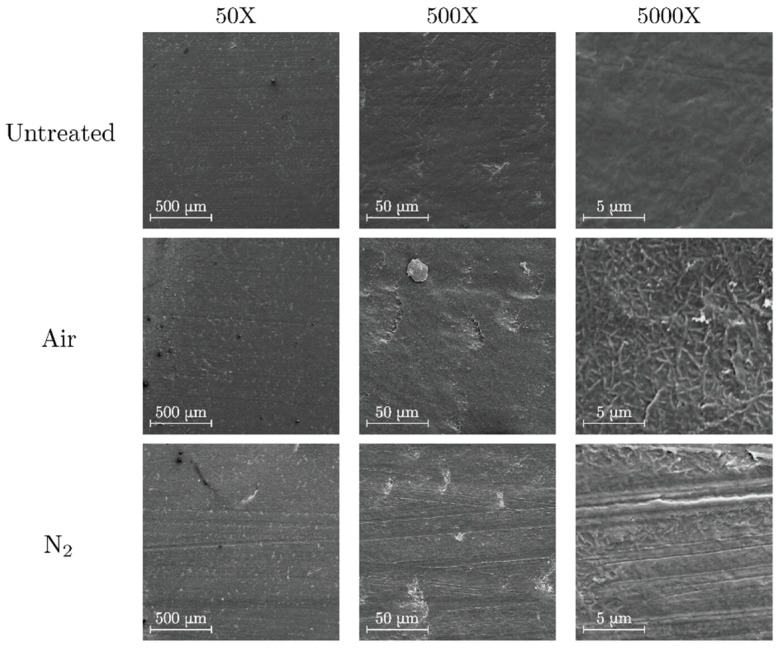
SEM images of untreated (**top**), air-treated (**center**), and nitrogen-treated (**bottom**) polytetrafluorethylene (PTFE).

**Figure 6 polymers-11-01646-f006:**
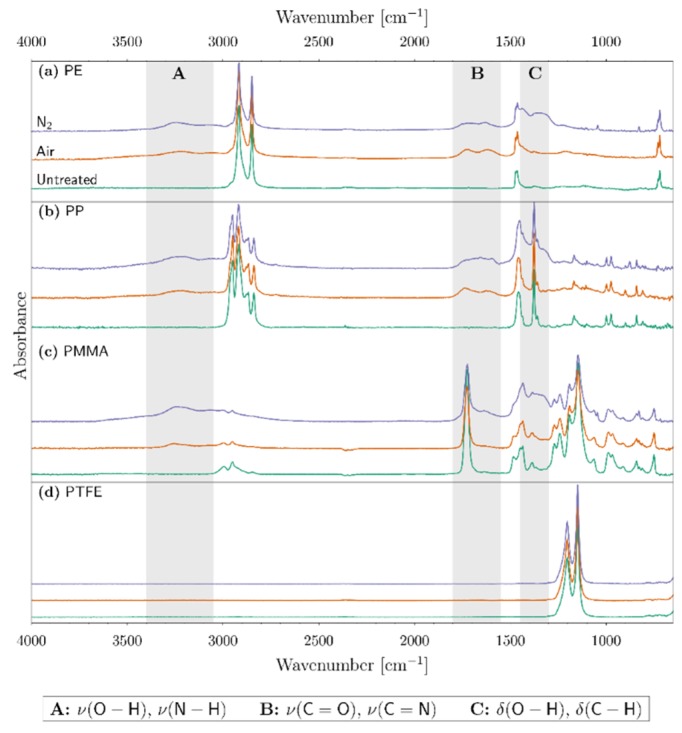
Infrared spectra of treated and untreated (**a**) PE, (**b**) polypropylene (PP), (**c**) PMMA, and (**d**) PTFE.

**Figure 7 polymers-11-01646-f007:**
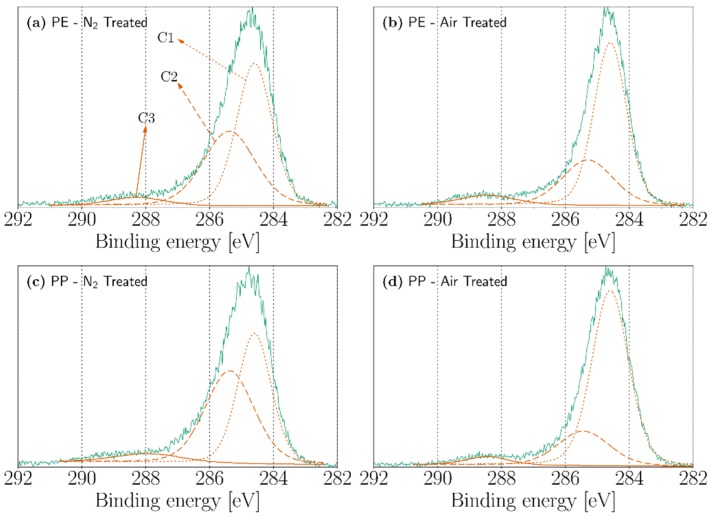
X-ray photoelectron spectroscopy (XPS) high-resolution carbon (C1s) spectra of (**a**) nitrogen-treated PE, (**b**) air-treated PE, (**c**) nitrogen-treated PP, and (**d**) air-treated PP.

**Figure 8 polymers-11-01646-f008:**
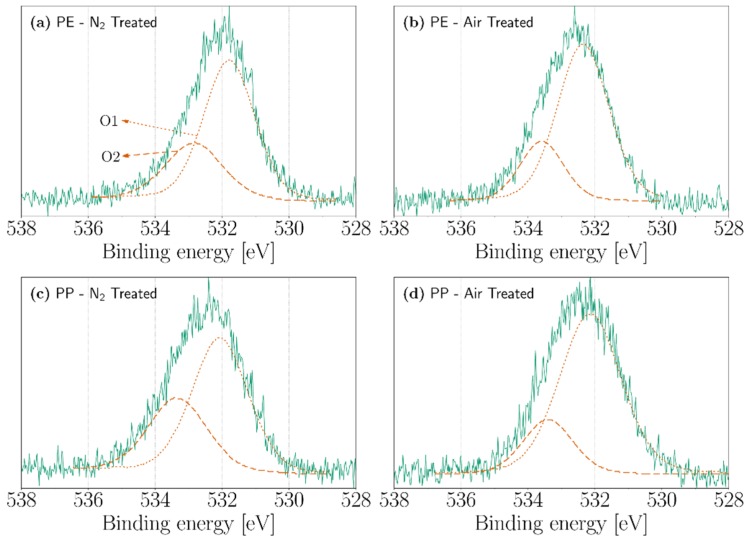
XPS high-resolution oxygen (O1s) spectra of (**a**) nitrogen-treated PE, (**b**) air-treated PE, (**c**) nitrogen-treated PP, and (**d**) air-treated PP.

**Figure 9 polymers-11-01646-f009:**
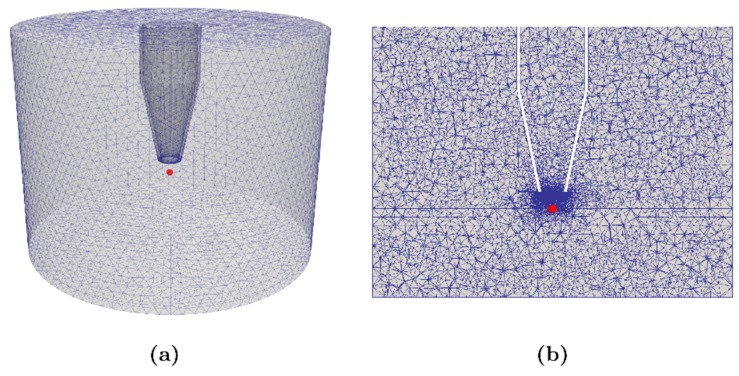
Computational domain of the experimental setup: the complete geometry (**a**) and a cross section (**b**).

**Figure 10 polymers-11-01646-f010:**
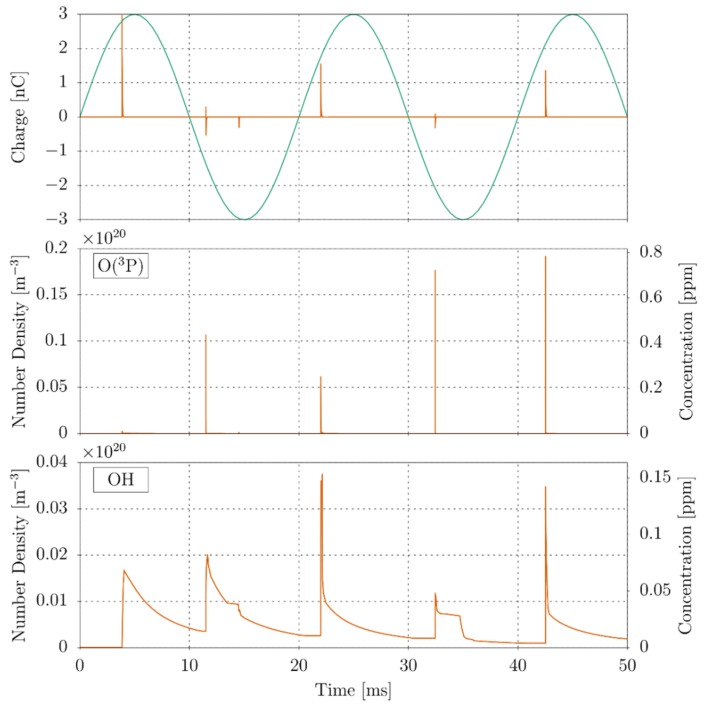
Temporal evolution of the electric current (**top**), oxygen atom (**middle**), and hydroxyl radical (**bottom**).

**Table 1 polymers-11-01646-t001:** Elemental composition (atomic %) of untreated and treated PE and PP surfaces.

Polymer	Treatment Atmosphere	Elemental Composition (%)			O/C Ratio	N/C Ratio
		C 1s	O 1s	N 1s		
Polyethylene	Untreated	92.47	7.53	−	0.08	−
	Nitrogen-treated	86.58	10.68	2.74	0.12	0.03
	Air-treated	83.21	16.06	0.73	0.19	0.01
Polypropylene	Untreated	93.14	6.86	−	0.07	−
	Nitrogen-treated	84.96	11.24	3.79	0.13	0.04
	Air-treated	88.47	10.62	0.91	0.12	0.01

**Table 2 polymers-11-01646-t002:** Relative peak concentration of the most abundant species.

Species	Relative Peak Concentration at Pulse (ppm)					
	1st (3.85 ms)	2nd (11.5 ms)	3rd (14.5 ms)	4th (22 ms)	5th (32.5 ms)	6th (42.5 ms)
O_3_	0.1134	3.042	2.414	3.945	1.842	3.38
O(^3^P)	0.0124	0.4374	0.0002	0.2527	0.7235	0.786
NO	0.0132	0.2182	0.1917	0.4287	0.1434	0.386
O_2_(a^1^Δ_g_)	0.0097	0.2842	0.2049	0.3498	0.157	0.2784
OH	0.0068	0.0826	0.0389	0.1528	0.0486	0.1427
N(^4^S)	0.0124	0.0483	0.014	0.2941	0.0179	0.3227

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
