# Peer review of "Experimental Characterization of Polymer Surfaces Subject to Corona Discharges in Controlled Atmospheres"

_polymers, 2019, doi:10.3390/polym11101646_

Round 1

Reviewer 1 Report

The authors presented a systematic study on the effect of corona discharge on various types of polymers. The study was well planned and executed. 

I have only one suggestion which is authors should compile a table of summary of their findings. 

If possible the authors should also compile another table of existing findings by other researchers in order to highlight their contribution.

Author Response

Answers to Reviewer-1

I have only one suggestion which is authors should compile a table of summary of their findings; If possible the authors should also compile another table of existing findings by other researchers in order to highlight their contribution.

We have restructured the conclusion section to better highlight the results and the comparison with the existing literature.

Reviewer 2 Report

General Comments

The manuscript entitled „Experimental characterization of polymer surfaces subject to corona discharges in controlled atmospheres” presents very interesting topic related with plasma treatment of polymers.

Specific Comments

The introduction focuses on plasma treatment of polymer surfaces. I think that extending the introduction by paragraph describing plasma polymerization will significantly enrich the reviewed paper. See Polymers 2018, 10, 532; doi:10.3390/polym10050532, Sensors 2018, 18, 4086; doi:10.3390/s18124086, Journal of Materials Science 2019, 54(15), 10746-10763 https://doi.org/10.1007/s10853-019-03641-2, Malinowski et al. doi: 978-1-5386-1943-8/17/$31.00 ©2017 IEEE The polymers were treated using the same nitrogen and air flow rate? If yes, the value should be written in section 2.2 (Experimental setup). If yes, why the authors choose only one the gas flow rate value? If not, how the gas flow rate does influence on morphology, chemical composition and wettability of plasma treated polymers? Another very interesting issue is distance between electrodes and treated polymer surface. Why did the authors choose distance 2.3 mm? It should be explained in section 2.2 (Experimental setup) Why the authors treated polymers? Could the authors indicate possible applications of plasma treated PE, PP, PMMA and PTFE?  Why in section 3.1 the authors did not study influence of plasma on morphology of PTFE? Why in section 3.3 the authors did not present chemical composition of untreated, nitrogen-treated and air-treated PMMA and PTFE (Table 1)? Figure 7 (line 253) and Figure 5 (line 264) should be supplemented by XPS high resolution spectra of nitrogen-treated and air-treated PMMA and PTFE. I think that the paper should be supplemented by Atomic Force Microscopy to determine roughness change after plasma treatment. The roughness is very important factor strongly influenced on materials practical application. Moreover I reckon that wettability studies could give new insight into studies materials. In my opinion conclusions based both XPS analysis and contact angle measurements would significantly enrich the paper.  

Minor Comments

In lines 101, 247/248, 281/282, 284, 296/297, 309/310, 322 are missing references, the Authors has to complete it.  The FTIR spectra of untreated and plasma treated PP (Figure 4b), PMMA (Figure 4c) and PTFE (Figure 4d) should be described like in case of untreated and plasma treated PE (Figure 4a). In the FTIR spectra of PTFE (Figure 4d) should be selected region D correspond to symmetric vs(C-F) and vas(C-F) In the paper figures are numbered wrongly.

In my opinion the paper entitled „Experimental characterization of polymer surfaces subject to corona discharges in controlled atmospheres” is interesting and significant but some analysis should be added. Overall, I suggest publication of this paper in POLYMERS after consideration above points.

Author Response

Answers to Reviewer-2

I think that extending the introduction by paragraph describing plasma polymerization will significantly enrich the reviewed paper. See Polymers 2018, 10, 532; doi:10.3390/polym10050532, Sensors 2018, 18, 4086; doi:10.3390/s18124086, Journal of Materials Science 2019, 54(15), 10746-10763 https://doi.org/10.1007/s10853-019-03641-2, Malinowski et al. doi: 978-1-5386-1943-8/17/$31.00 ©2017 IEEE: the papers cited above have been included in the discussion; The polymers were treated using the same nitrogen and air flow rate? If yes, the value should be written in section 2.2 (Experimental setup). If yes, why the authors choose only one the gas flow rate value? If not, how the gas flow rate does influence on morphology, chemical composition and wettability of plasma treated polymers? Another very interesting issue is distance between electrodes and treated polymer surface: the samples are aged in a controlled atmosphere but no gas flow is imposed. The sequence of discharges is started only when the specific gas for the test is loaded in the test chamber and it reaches some proper pressure conditions. The test is conducted in a still gas no flow is present; Why did the authors choose distance 2.3 mm? It should be explained in section 2.2 (Experimental setup): we have added a remark in the paper; Why the authors treated polymers? Could the authors indicate possible applications of plasma treated PE, PP, PMMA and PTFE? We are not seeking a particular application for these treatments, rather we are interested in the effect of plasma-plastic interactions since these phenomena affect the evolution of the ageing of electrical insulation materials. We have detailed more precisely why we are analysing a number of polymeric materials; Why in section 3.1 the authors did not study influence of plasma on morphology of PTFE? Why in section 3.3 the authors did not present chemical composition of untreated, nitrogen-treated and air-treated PMMA and PTFE (Table 1)? Figure 7 (line 253) and Figure 5 (line 264) should be supplemented by XPS high resolution spectra of nitrogen-treated and air-treated PMMA and PTFE: XPS was not performed on PTFE since infrared spectroscopy essentially shows that no significant modifications were introduced after treatment with the discharge. On the other hand, PMMA possesses functional groups that already contain oxygen and would probably mask any contribution due to the treatment with the corona discharges. It is also worth noting that the main focus was put on PP and PE since they are interesting polymers in the power transmission industry. We have added a comment in the text; I think that the paper should be supplemented by Atomic Force Microscopy to determine roughness change after plasma treatment. The roughness is very important factor strongly influenced on materials practical application: we decided not to pursue AFM measurements since these would not provide any information regarding the chemical composition of the surface. The idea in the paper is to lay a bridge between corona discharges, subsequent creation of active species (even at low concentrations) and resultant chemical modifications of the surface that will lead to the ageing of the specimen. On the other hand, SEM images already provide an idea of the topological features that appear after the treatment; Moreover, I reckon that wettability studies could give new insight into studies materials. In my opinion, conclusions based both XPS analysis and contact angle measurements would significantly enrich the paper: the reason we did not perform wettability studies on treated polymers is related to what was said in the previous point. Since we have a particular interest in the chemical modifications taking place on polymer surfaces, contact angle measurements were not considered at first since they would not provide further chemical composition insight in comparison with XPS and FTIR. Moreover, a water drop is relatively large compared to the aged area of the plastic samples. The contact angle changes with respect to the zone where the drop is placed. Thus, much larger areas are needed to get reliable data; In lines 101, 247/248, 281/282, 284, 296/297, 309/310, 322 are missing references, the Authors has to complete it: fixed; The FTIR spectra of untreated and plasma treated PP (Figure 4b), PMMA (Figure 4c) and PTFE (Figure 4d) should be described like in case of untreated and plasma treated PE (Figure 4a): the explanation of the fundamental contributions of untreated polymers was already included in the lines 108-288, however, we realize that this may not be clear for the reader so we have added two subsections: 3.2.1 for untreated polymers and 3.2.2 for treated polymers; In the FTIR spectra of PTFE (Figure 4d) should be selected region D correspond to symmetric v (C-F) and v (C-F): the purpose of the three shaded areas (marked as A, B and C) shown in the FTIR spectra was to indicate the regions where the most relevant contributions appear after the plasma treatment in most of the polymers (except for PTFE that remains unaltered). Thus, their purpose is not to show the contributions of the functional groups already present in untreated polymers. We have kept the figure as it is, but we have provided a clearer explanation in the text; In the paper figures are numbered wrongly: fixed;

Reviewer 3 Report

This research seems solidly performed, and well documented.  There are a few issues with the manuscript as prepared, namely a large number of defective references which are seemingly to the figures within the paper itself which need to be corrected.

In terms of scientific content, a few comments, observations, and/or requests:

1)  Much time is spent discussing the fact that discharge under a positive nitrogen atmosphere nevertheless leads to oxygenation of the defect surfaces, yet this explanation is left to "presuming" the chemical reactions leading to conversion.  Since the work is already done under an external atmosphere, could the outgassing products not be collected and quantified.  Surely the various amino products from exchange reactions would be detectable?

2)  A bit more description of the modeling of the corona discharge is in order.  It's not necessary to completely and thoroughly describe the model, but at the very least the process the model simulates, the way that it does so, and the way this feeds the modeling output data should be described.  From context I presume that the model simulates the charge accumulation on the electrode tip as well as the quiescent charge simulation/breakdown kinetics in the continuum, leading to random discharge events as temporary ion pathways allow for conduction of the charge as potential builds.  This would e.g. be necessary for the seemingly randomized discharge times listed in the table.  I shouldn't be expected to go and look up the reference for this simple description, however, it should be provided, and then readers interested in more specifics may be referenced to the necessary works.

3)  I'm having trouble following the argument in lines 328-333.  It's noted that the concentration of O(3P) is about twice that of OH, and this means that the polymer degradation will be "dominated" by the O(3P) pathway in Eq. 4.  However, the reaction probability of O(3P) for hydrogen extraction is given as 500 times more likely for OH than O(3P), and eqns. 4/5 seem to have kinetics that are linear in the reactive species.  I don't understand the basis for the argument being made that O(3P) dominates the chemistry.  I also wonder about using the time average ratios, when Figure 10 demonstrates that in some cases the reactant concentration is heavily dominated by one species or the other.

4)  Related to 3 above, if we believe the time evolution of the modelled reaction from Figure 10, why do we see a slow decay in OH vs. the peak and subsequent immediate removal of O(3P)?  This would seem inconsistent (and in fact inverted) with the noted reactivity referenced in #3 above.  How does the reaction model account for the formation of OH by the O(3P) reaction?

Overall this work seems solid and should be acceptable, but the modeling portion seems to have been included in a "black box" type of fashion and more exposition/analysis of the model is needed in order to be convincing that there are actual connections between the observed phenomenon and the model being used to understand it.

Round 2

Reviewer 3 Report

The additional description and explanation of the modeling and kinetic section significantly clarify both the work and results.  The article is solidly written and analyzed, and clearly ready for publication.